# Humanity's diverse predatory niche and its ecological consequences

Chris T. Darimont [1,2,15✉], Rob Cooke [3,15✉], Mathieu L. Bourbonnais[4], Heather M. Bryan[2,5],
Stephanie M. Carlson [6], James A. Estes[7], Mauro Galetti[8,9], Taal Levi[10], Jessica L. MacLean [1,2],
Iain McKechnie [11,12], Paul C. Paquet[1,2] & Boris Worm[13,14]

Although humans have long been predators with enduring nutritive and cultural relationships with their prey, seldom have conservation ecologists considered the divergent predatory behavior of contemporary, industrialized humans. Recognizing that the number, strength and diversity of predator-prey relationships can profoundly influence biodiversity, here we analyze humanity's modern day predatory interactions with vertebrates and estimate their ecological consequences. Analysing IUCN 'use and trade' data for ~47,000 species, we show that fishers, hunters and other animal collectors prey on more than a third (~15,000 species) of Earth's vertebrates. Assessed over equivalent ranges, humans exploit up to 300 times more species than comparable non-human predators. Exploitation for the pet trade, medicine, and other uses now affects almost as many species as those targeted for food consumption, and almost 40% of exploited species are threatened by human use. Trait space analyses show that birds and mammals threatened by exploitation occupy a disproportionally large and unique region of ecological trait space, now at risk of loss. These patterns suggest far more species are subject to human-imposed ecological (e.g., landscapes of fear) and evolutionary (e.g., harvest selection) processes than previously considered. Moreover, continued over-exploitation will likely bear profound consequences for biodiversity and ecosystem function.

[1] Department of Geography, University of Victoria, Victoria, BC, Canada. [2] Raincoast Conservation Foundation, Sidney, BC, Canada. [3] UK Centre for Ecology & Hydrology, Wallingford, UK. [4] Department of Earth, Environmental, and Geographic Sciences, University of British Columbia Okanagan, Kelowna, BC, Canada. [5] Department of Ecosystem Science and Management, University of Northern British Columbia, Prince George, BC, Canada. [6] Department of Environmental Science, Policy, and Management, University of California, Berkeley, CA, USA. [7] Department of Ecology and Evolutionary Biology, University of California, Santa Cruz, CA, USA. [8] São Paulo State University (UNESP), Department of Biodiversity, Rio Claro, São Paulo, Brazil. [9] Kimberly Green Latin American and Caribbean Center, Florida International University (FIU), Miami, FL, USA. [10] Department of Fisheries, Wildlife, and Conservation Sciences, Oregon State University, Corvallis, OR, USA. [11] Department of Anthropology, University of Victoria, Victoria, BC, Canada. [12] Hakai Institute, Heriot Bay, Quadra Island, BC, Canada. [13] Department of Biology, Dalhousie University, Halifax, NS, Canada. [14] Ocean Frontier Institute, Dalhousie University, Halifax, NS, Canada. [15] These authors contributed equally: Chris T. Darimont, Rob Cooke. ✉email: darimont@uvic.ca; RobOke@ceh.ac.uk

Predation evolved as an ecological strategy to acquire energy and nutrients from a limited number of species vulnerable to attack. The strength and diversity of these predatory interactions often exert strong influence on the structure and functioning of ecosystems. For example, predation can affect the diversity, abundance, and evolution of co-existing species, energy flows, and disease dynamics[1–6]. The planet's predatory landscape has long included *Homo sapiens*[7]. Yet, with the rise of advanced hunting and fishing technology, global commercialization, trade, and more, interactions between people and their prey have changed profoundly.

Here we estimate the extensive interactions that now link contemporary human predators to other vertebrates by quantifying humanity's predatory niche. We also: (i) ask how these interactions might threaten prey species, (ii) compare humanity's predatory niche with other widespread predators, and (iii) draw on ecological trait data of terrestrial bird and mammalian prey to identify the potential outcomes of losing overexploited species in terms of the ecological diversity present in ecosystems.

We approach these questions in new ways. Recent work on the sustainability of exploitation has focused on specific taxa[8], uses[9], and areas of intense exploitation[10], estimating impacts in the context of extinction risk according to the International Union for the Conservation of Nature (IUCN). In contrast, our analysis spans the planet's vertebrates and all forms of predation, and predicts the potential aggregate ecological consequences of overexploitation. We assess the predatory niche of all contemporary humans—collectively—using a 'snapshot' of contemporary data. As conservation scientists, we focus on the potential ecological harms associated with overexploitation. We also compare our results with a summary of contemporary subsistence use, as well as discuss more broadly the enduring legacies—and conservation promise—of place-based management systems that have enabled people to exploit species sustainably over millennia.

We consider predation by humans broadly—and from the perspective of effects on prey populations—as any use that removes individuals from wild populations, lethally or otherwise. Processes we considered (as captured by IUCN 'use and trade' designations; below) ranged from removal of live individuals for the pet trade, to harvesting by societies that rely heavily on hunting and fishing, to globalized, commercial fishing and trade of vertebrates, and interactions among these activities. We additionally considered this broad definition by reasoning that these varied activities all include processes (i.e., detection, capture, etc.) exemplified by predation.

To capture this breadth of activities, we collated IUCN 'use and trade' data, which categorize the uses for which species are killed (e.g., for human food, animal feed, sport hunting/specimen collection) or collected from the wild (e.g., for use as pets, for establishing ex-situ production) for 46,755 vertebrate species; across the six vertebrate classes with the most species (i.e., excluding classes with <100 species), we also examined IUCN assessment data for every listed species (i.e., threats, Red List status; see *Describing Uses & Threats* in Methods). IUCN Red List data are ultimately derived from expert knowledge, in combination with empirical data where available. Here, we consider assessments of species that both: (i) identify exploitation as a threat and (ii) list them as at risk of extinction (i.e., Vulnerable, Endangered, or Critically Endangered IUCN Red list status) as a signal of overexploitation. We find that humans exploit almost 15,000 vertebrate species for diverse food and non-food uses, and endanger many species—and their roles in ecosystems—via overexploitation.

## Results

### Human use of vertebrates and extinction risk. Humans use roughly a third of the species across the six vertebrate classes we

examined ($n = 14,663$ of 46,755; 31%; Fig. 1a). Of these exploited species, only about 55% (8,037 species; 17% of total species assessed) are killed for food (see below for patterns related to other uses). Human prey diversity is highest across marine prey species (43% of assessed taxa), followed by freshwater (35%) and terrestrial (26%) species. Almost half of all ray-finned fishes (42%) and birds (46%) are used by humans, accounting for 11,697 (78%) of all exploited vertebrates. Mammals and cartilaginous fishes show intermediate use (24% and 28% of species, respectively), whereas reptiles and amphibians are least exploited (14% and 8%, respectively; Fig. 1a).

How exploitation relates to extinction risk varied among taxa. Human use is considered a threat for 12% of all vertebrates ($n = 5775$ of 46,755) and 39% of used vertebrates ($n = 5775$ of 14,663; Fig. 1b). Moreover, 4% of all vertebrates and 13% of used vertebrates that face extinction (classified as Vulnerable, Endangered, or Critically Endangered) have human use recognized as a threat ($n = 1859$; Fig. 1b). The extent to which human use contributes to extinction risk ranges from 6% in exploited ray-finned fishes to 36% in exploited mammals (Fig. 1b).

Compared with other wide-ranging predators of vertebrates (i.e., predatory fishes, sharks, avian and mammalian predators), humans exploit many more vertebrate species. Paired comparisons over equivalent geographic ranges with 19 vertebrate predators for which range-wide dietary data exist (see *Prey diversity comparisons* in Methods) revealed that humans exploit ~5 to ~300 times the number of vertebrate species (~4 to ~122 times, considering food items only; Fig. 1c). Prey overlap between humans and these predators is also pronounced, ranging from 30% (of Bigeye Tuna prey) to 100% (Jaguar; Fig. 1c), with a median of 69%.

### Diversity of uses of vertebrates by humans. This extraordinarily large predatory niche reflects a striking diversity of uses. IUCN data identify 18 categories, ranging from food for humans, pets (i.e., companion or captive animals) and sport hunting/collection to less common uses like clothes, medicines, animal feed, and poisons (Fig. 2; Supplementary Table 1). Multiple uses for individual species are common (26% of species). Food use (i.e., eaten by humans) dominates the exploitation of marine and freshwater fishes (72% of species). By contrast, in the terrestrial realm, use as pets is almost twice as common (74%) as food use (39%). Sport hunting and other forms of collection (i.e., for trophies and ornaments, etc.) underlies use of 8% of exploited terrestrial species (Fig. 2). Taxonomic patterns show that fishes and mammals are mostly used for food, whereas birds, reptiles, and amphibians are primarily targeted as pets (Fig. 2).

### Geographical patterns of use of vertebrates by humans. Using available range maps from the IUCN and BirdLife International (see *Mapping* in Methods), we documented pronounced geographic variation in human use of vertebrates. Equatorial regions, where species richness is highest, particularly coastal areas and across Southeast Asia, show the highest number of exploited species (Fig. 3a). Standardizing for species richness revealed areas of disproportionately high (e.g., most ocean basins; India, North Africa, Northern Eurasia) or low (Southern Ocean; Central and South America; South and Eastern North America) use of species (Fig. 3b). Food is the most common use of vertebrates across the oceans, and on land across Eurasia and in Southeast Asia (Fig. 3c). Pet use accounts for more than half of exploited species across most of the terrestrial regions of the planet as well as in marine areas surrounding archipelagos (Fig. 3d).

### Ecological trait space analyses. To provide additional conservation context, which relates to ecosystem function, we also estimated

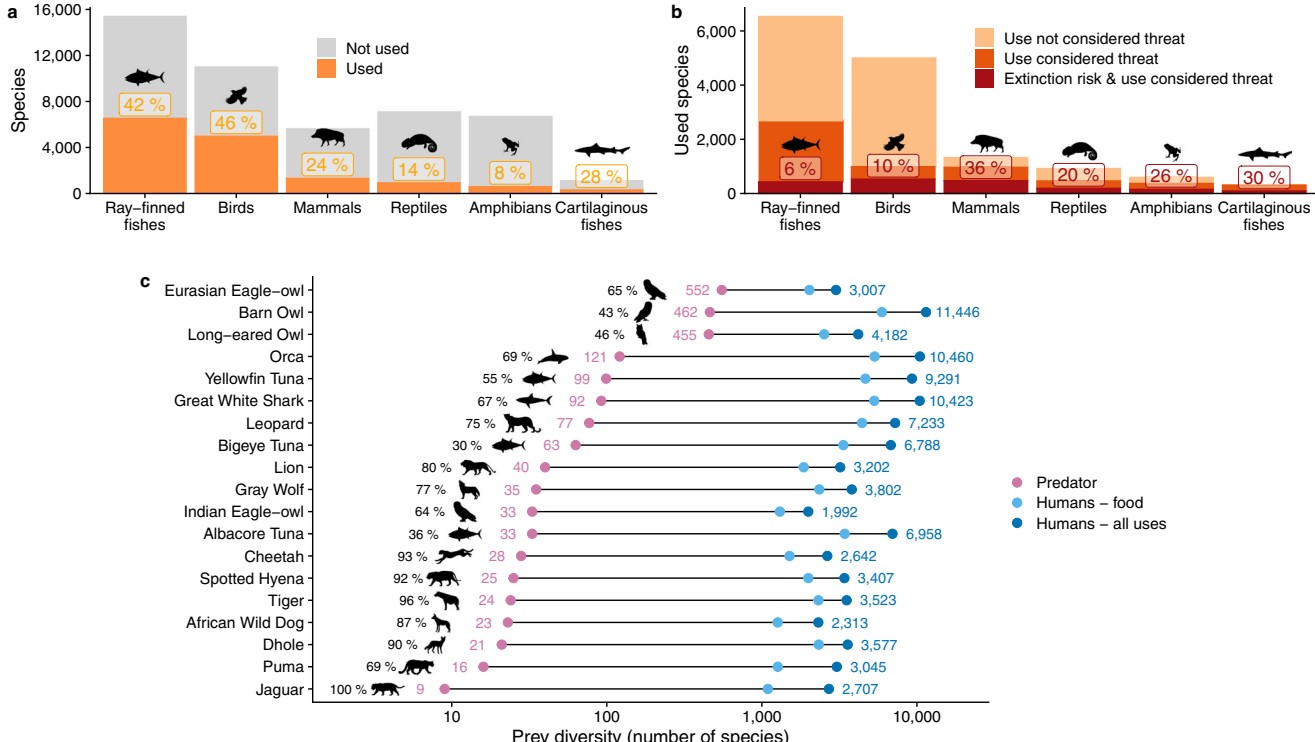

**Fig. 1 Use of vertebrates by humans and other predators. a** Number and percent of vertebrate species with documented human use, and **b** number for which use is considered a threat, including the subset facing extinction (Vulnerable, Endangered, or Critically Endangered status on the IUCN Red list). **c** Prey diversity (number of species; logarithmic scale) of humans and comparable predators (i.e., those that prey on vertebrates for which range-wide data were available) across equivalent geographic ranges, with percentages indicating human prey overlap with each predator.

the patterns and consequences of exploitation on the diversity of ecological traits across terrestrial birds and mammals ($n = 16,413$ species). To do so we collated data for five traits (body mass, generation length, diet, habitat breadth, litter/clutch size), imputing missing values where necessary; we then used principal component analysis to examine if humans exploit (and overexploit) species non-randomly across trait space, assessing the volume and uniqueness of exploited trait space (details in *Trait data* to *Ecological trait space* sections of Methods). We found that humans target species that are larger-bodied, longer-lived, have more herbivorous diets, and have larger habitat breadths than those species not used (Fig. 4a; Supplementary Fig. 1). Moreover, those species at risk of extinction and for which use is considered a threat occupy a disproportionally large (Fig. 4b; permutation test $P = 0.01$) and unique (Fig. 4c; $P = 0.01$) region of trait space. Finally, humans are seemingly unique among predators in interacting so broadly with the ecological trait space of birds and mammals; paired comparisons with the same group of widely ranging vertebrate predators used in prey diversity comparisons (Fig. 1c) show that humans exploit volumes of trait space that are ~1.2 to ~1300 times greater (Fig. 4d).

## Discussion

Our comprehensive assessment revealed an unparalleled taxonomic, spatial, and ecological breadth of humanity's predatory niche. This uniquely large predatory role is up to 300 times taxonomically and 1300 times ecologically larger than those of the non-human predators to which we had comparable data, and is driven by a wide variety of uses, many of which are independent of sustenance. Use for pets, medicines, and other wildlife products, for example, are not only common (Figs. 2; 3d) but also now pose a key threat to endangered wildlife in many areas[9].

Moreover, our assessment is seemingly conservative; for example, approaches using different methods and taxonomic resolutions have estimated higher proportions of endangered taxa among amphibians[11] and reptiles[12]. We also note that our contemporary 'snapshot' of IUCN assessments cannot capture the exploitation-related loss of species (i.e., 'defaunation') that has already occurred over previous centuries[13] and millennia[14] of predation by humans. On the other hand, despite a high proportion and enormous number of species considered by the IUCN as threatened by exploitation, for many vertebrates (e.g., most ray-finned fishes) harvests are not considered a threat to populations (Fig. 1b). Moreover, many species can face more severe threats from other human activities, namely habitat destruction, invasive species, and climate change. Notably, however, in a recent, taxonomically broad and global assessment, Jaureguiberry et al.[15] found that direct exploitation and land/sea use change were dominant drivers of biodiversity loss.

How did such an extraordinarily large predatory niche evolve? An evolutionary perspective would highlight associations among meat-eating, advanced cognition, and tool use[7]. Cognitive and cooperative hunting abilities unparalleled among predators enabled the development of sophisticated technology, from stone-crafted projectiles to fossil-fuel powered vehicles equipped with sensitive prey detection equipment[16]. Such advances allowed humans to escape the limitations of foraging over finite space and to be able to encounter novel prey and overcome—or even capitalize on—evolved anti-predator defenses[17,18]. Technologically advanced fishing vessels and their gear, for example, have allowed a 'terrestrial organism' to become a highly efficient marine predator on the open ocean with nets that take advantage of the otherwise adaptive schooling behavior of many fishes. Moreover, as overexploited species collapse, new ones are targeted[19]. An anthropological view would also invoke humanity's well-developed material, medicinal,

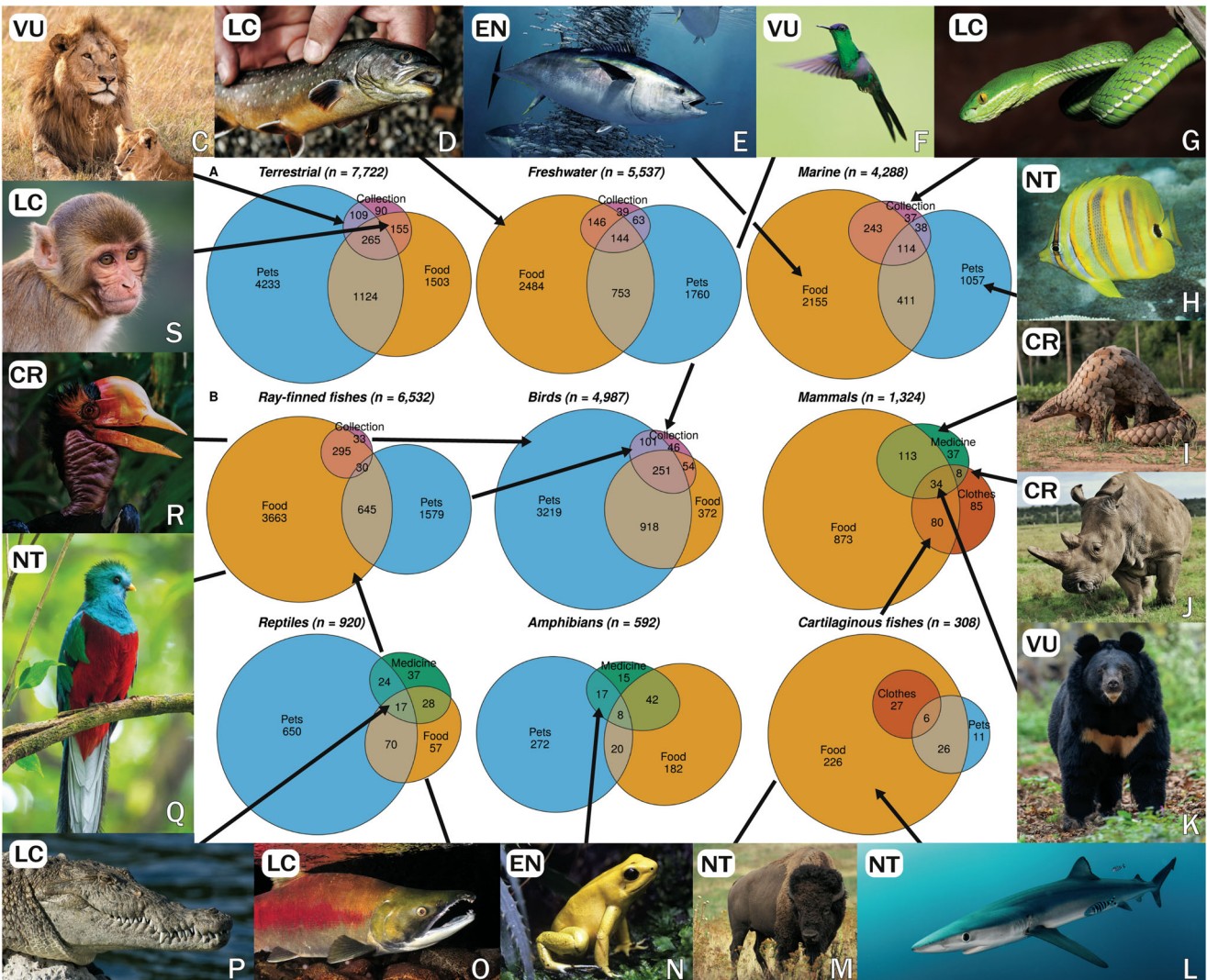

**Fig. 2 Diversity of uses by human predators.** Number and overlap of species in each IUCN 'use and trade' category for **A** terrestrial and aquatic realms and **B** six vertebrate classes with the most species. Images depict examples of exploited species in use categories along with their IUCN status (LC: Least Concern, NT: Near Threatened, VU: Vulnerable, EN: Endangered, CR: Critically Endangered). Taxonomic information available in Supplementary Information (Supplementary Table 2). **C** African lion, *Panthera leo* (photo: Antony Trivet via Pixabay). **D** Arctic char, *Salvelinus alpinus* (photo: Reinhard Thrainer via Pixabay). **E** Atlantic Bluefin Tuna, *Thunnus thynnus* (photo: Marko Steffensen via Alamy). **F** Violet-capped Woodnymph, *Thalurania glaucopis* (photo: Wilfred Marissen via iStock). **G** White-lipped viper, *Trimeresurus albolabris* (photo: Mark Kostich via iStock). **H** Rainford's butterflyfish, *Chaetodon rainfordi* (photo: Biosphoto via Alamy). **I** Philippine pangolin, *Manis culionensis* (photo: Vicky Chauhan via iStock). **J** Northern rhinoceros, *Ceratotherium simum cottoni* (photo: Adele Dobler via iStock). **K** Asiatic black bear, *Ursus thibetanus* (photo: Volodymyr Burdiak via Shutterstock). **L** Blue shark, *Prionace glauca* (photo: Howard Chen via iStock). **M** American bison, *Bison bison* (photo: WikiImages via Pixabay). **N** Golden poison frog, *Phyllobates terribilis* (photo: Hippopx.com). **O** Sockeye salmon, *Oncorhynchus nerka* (photo: Eduardo Baena via iStock). **P** American crocodile, *Crocodylus acutus* (photo: Pixabay). **Q** Resplendent quetzal, *Pharomachrus mocinno* (photo: Mikhail Dudarev via iStock). **R** Helmeted hornbill, *Rhinoplax vigil* (photo: Craig Ansibin via Shutterstock). **S** Rhesus macaque, *Macaca mulatta* (photo: Donyanedomam via iStock.com).

and companion animal culture[20,21] that contributes to diverse non-food uses. Ecologically, however, these uses have the same effect as predation for food by removing individuals from populations. Finally, spatial patterns of vertebrate capture (Fig. 3) belie the reality that many animals are consumed far from their regions of provenance[22]. In this way, global commerce and trade—uniquely human endeavors—underlie the industrialization of humanity's relationships with many species across its diverse predatory niche.

Although detailed comparisons at standardized scales are not possible, humanity now likely has a far broader predatory niche than at any time in history. On one hand, at the end of the Pleistocene, human prey diversity contracted with numerous megafaunal extinctions[23]. On the other hand, the early

diversification of the human niche proceeded with and paralleled the development of environmental management techniques (e.g., fire) and the later advent of agriculture and animal husbandry[7], a process that Flannery[24] termed the 'broad spectrum revolution'. Contemporary subsistence peoples, however, also show clear hunting preferences that reflect long-entwined history of interaction with specific environments[25,26]; a global meta-analysis, drawing on ~800,000 kill records, identified 504 vertebrate prey species (~3.3% of the prey diversity we detected among all human predators), as well as evidence for avoidance of smaller prey and a preference for a small number of larger-bodied animals[25].

The rise of the planet's most widespread predator affects an enormous global network of interaction chains connecting

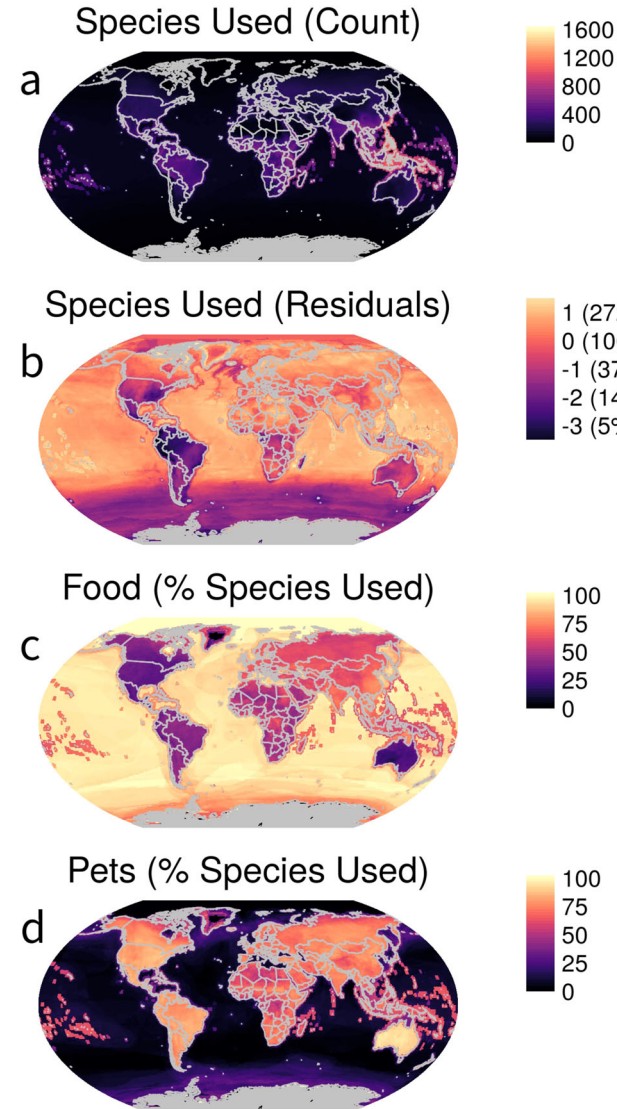

**Fig. 3 Spatial patterns of vertebrate use by human predators. a** Number of species used. **b** Number of species used after accounting for variation in species richness (standardization process described in *Mapping* section of Methods; Fig. 5). Percent of used species that are exploited as **c** food and **d** pets. Patterns relate to the distribution of species (assessed across their entire range), not necessarily where capture, consumption or other end use occurs.

humans to their prey, with many direct and indirect consequences. For example, our data on prey overlap suggests humans are not only generalist predators but also might compete strongly with other predators. Moreover, owing to divergent phenotypic targets compared with other predators, hunters and fishers can exert rapid phenotypic and evolutionary changes in their prey[27,28]. Against this background of a broad niche, high mortality risk, and strong selection pressures, even the perceived threat of predation associated with benign human activity has altered the behavior of many taxa[29–31]. Should exploitation be as taxonomically widespread as the patterns we present suggest, and exploitation rates as consistently high as previous analyses have suggested[18,32], humanity's predatory niche likely affects a much larger suite of species, areas, and processes than currently identified, including the ecology of fear[33] and harvest selection[27,28].

Although affecting only a moderate proportion of all vertebrates used, the large number of species for which exploitation is considered a threat might likely contribute to continued loss of species (Fig. 1b; those species facing extinction risk), as well as loss of variation in ecological trait space (Fig. 4a–c). Without changes to predatory behavior by humans, these losses are likely to further reduce the ecological diversity present among the world's vertebrates[34] (Fig. 4), with consequences for global ecosystem functioning[35,36]. Taxonomic losses might expand as species subject to intense exploitation decline or earn protections, causing hunters, fishers, and collectors to switch to other species that are phylogenetically, morphologically, and ecologically similar[9].

Confronting the potential loss of species and the associated variation in ecological strategies present in ecosystems requires an interdisciplinary and inclusive approach that recognizes historic and enduring interactions between Indigenous, place-based societies and prey with which they have maintained relationships over millennia. Collaborations among social and natural scientists, as well as conservation practitioners, have looked to these interactions to learn about how social and cultural practices can mitigate humanity's tendency to overexploit prey populations over time[37]. As one example, oral histories and archeological data provide compelling evidence that place-based practices of Indigenous stewardship supported sustainable harvests of Pacific herring (*Clupea pallasii*) over millennia before industrial over-exploitation caused rapid population collapses[38]. Instructive case studies like this that illustrate the cultural and place-based underpinnings of decentralized harvest management provide important contrasts to the often centralized 'command-and-control' approaches used in industrial exploitation. Notably, restoration of decentralized governance systems of harvesting and the sustainability benefits they can manifest[38] align with global aspirations towards social justice, as codified in the United Nations Sustainability Development Goals and Declaration on the Rights of Indigenous Peoples, among others. Regardless of conservation approach, we suggest more broadly that society needs to fully recognize the comprehensive effects that humanity's out-sized predatory niche can exert not only on target species but also their ecosystems. Although humanity's predatory niche is seemingly unrestricted, exploitation rates need to be constrained if >45,000 contemporary vertebrate species and the ecological processes they support are to be safeguarded.

## Methods

**Data acquisition**. We downloaded taxonomic names, use and trade information, as well as threat and status data from the IUCN Red List (iucnredlist.org[39]) by apiV4 in April 2019. We used global IUCN assessments, excluding regional assessments. We acknowledge that although empirical evidence often underlies IUCN assessments, they are completed by people with privileged access to the evaluation process who hold particular values, cultures, and ideological commitments. Moreover, those who assess do not declare whether they hunt or fish the species they assess.

We focused on contemporary large-scale patterns of human use among the largest groups of vertebrates. We excluded vertebrate classes with <100 species, which left 6 major vertebrate classes (Actinopterygii [Ray-finned fishes], Aves [Birds], Reptilia [Reptiles], Amphibia [Amphibians], Mammalia [Mammals], and Chondrichthyes [Cartilaginous fishes]). Data were summarized only for extant species by excluding species listed as Extinct (EX) or Extinct in the Wild (EW).

Despite the broad definition we have used, our estimate of humanity's predatory niche is likely conservative for several reasons. For example, not all known (or unknown) vertebrate species have Red List assessments (or are assessed by us; above). Additionally, there might be some biases among vertebrate classes; for example, coverage for reptiles and marine/freshwater taxa is less than for other vertebrates[40]. Moreover, for 1,499 vertebrates IUCN lists 'biological resource use' (i.e., including exploitation) as a threat but for which use and trade data are absent (i.e., these species either have incomplete use and trade data, or are threatened indirectly by the use of interacting species, e.g., from logging or bycatch [we excluded species if their only use was via bycatch]). Our comparison with CITES records of vertebrates at the species level returned an additional 600 species with no use data in IUCN[40]. We did not incorporate these additional 600 records into our dataset, so that our analyses would remain consistent and comparable across taxa under the IUCN framework (i.e., avoiding differences in taxonomy and differences

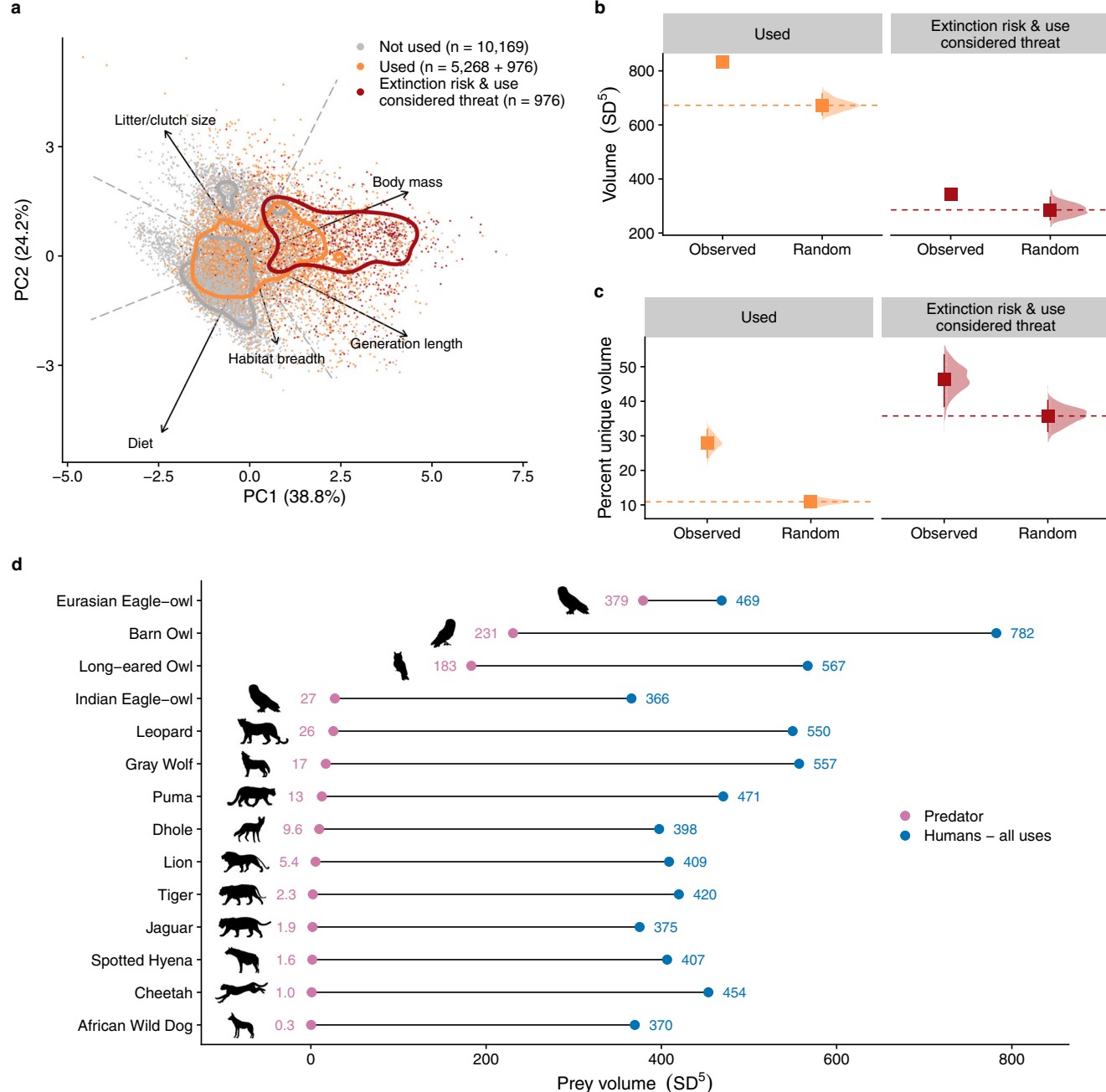

**Fig. 4 Human use of birds and mammals across trait space. a** Position of 16,413 terrestrial bird and mammal species across ecological and morphological trait space—colored by use, threat and extinction risk; contours indicate 50% probability contours. Comparisons of **b** total volume and **c** unique volume between observed and randomized trait spaces; squares indicate medians, and bars indicate 95% confidence intervals. 'Extinction risk' refers to species assessed as Vulnerable, Endangered, or Critically Endangered by the IUCN. **d** Prey volume of comparable predators and humans across equivalent geographic ranges. Only 'Human-all uses' is visualized; estimates for 'Human-food' are approximately equivalent, due to the non-linear scaling of volume with the number of species. See *Ecological trait space* section of Methods for details.

in the definition of 'use and trade'). In addition, previous analyses, using differing methodologies and taxonomies, have identified greater use/trade for some taxonomic groups than we observed[11,12]. For instance, Marshall et al.[12] found that 36% of reptile species are traded, based on data from a combination of web-based private commercial trade, CITES and the U.S. Fish and Wildlife Service Law Enforcement Management Information System (LEMIS), following the Reptile Database taxonomic framework. In contrast, we found that 14% of reptiles are used based on IUCN data and the IUCN taxonomic framework. Thus, depending on taxonomic differences, our results are likely conservative and might miss unregulated, illicit, or undetected trade; highlighting that trade may be affecting far more species than those actively monitored and assessed[11,12]. Furthermore, we did not estimate exploitation of invertebrates, which are commonly and increasingly exploited[41]. Finally, our estimate provided only a snapshot of use in contemporary

periods; on one hand, human-animal relationships continue to unravel with the loss of subsistence societies[42], but in other contexts de novo exploitation of species has accelerated[19] or is predicted to do so[9].

**Describing uses & threats.** We defined use by humans based on the 'Use and Trade' section of IUCN species assessments (Version 1.0; https://www.iucnredlist.org/resources/general-use-trade-classification-scheme; Supplementary Table 1). We classified species as 'Not used' ($n = 32,092$ species), where the species had no documented use, and 'Used' ($n = 14,663$ species) where the species had a documented use. We note that some use and trade categories encompass multiple uses for which we use abbreviated labels in the manuscript (e.g., 'Pets' for those categorized under 'Pets/display animals, horticulture' use; Supplementary Table 1). We

also extracted threat types from IUCN assessments, where threats from use are those listed under the 'Biological Resource Use' category[8]. These included threats identified by the following IUCN threat codes (Version 3.2; https://www.iucnredlist.org/resources/threat-classification-scheme): 5.1\$, 5.1.1, 5.1.2, 5.1.3, 5.1.4, 5.4\$, 5.4.1, 5.4.2, 5.4.3, 5.4.4, 5.4.5, and 5.4.6. Using these use and threat data, we further divided species used by humans (14,663 species) into 'Use not considered threat'—species without biological resource use listed as a threat ($n = 7029$ species), 'Use considered threat'—non-threatened species with biological resource use listed as a threat ($n = 5775$ species), and 'Extinction risk & use considered threat'— threatened (i.e., Vulnerable [VU], Endangered [EN], Critically Endangered [CR]) species with biological resource use listed as a threat ($n = 1859$ species).

We also compared use across realms. We assigned species to all realms that they inhabit—terrestrial ($n = 29,917$ species), freshwater ($n = 15,823$ species), marine ($n = 9912$ species)—based on IUCN designations. For instance, European Common Frog, *Rana temporaria*, was assigned to both the terrestrial and freshwater realms, reflecting the fact that they could be used in both these realms.

To visualize the use data we generated area-proportional Euler diagrams (a generalization of a Venn diagram; Fig. 2), which display proportions of, and overlaps between, use categories (Supplementary Table 1), with circles/ellipses. We fit the Euler diagrams with the *euler* function, which uses numerical optimization to find exact or approximate solutions to display proportions and overlaps. We grouped species based on their realms (terrestrial, freshwater, marine) and taxonomic classes (ray-finned fishes, birds, reptiles, amphibians, mammals, and cartilaginous fishes).

**Prey diversity comparisons**. We identified a geographically and taxonomically diverse set of comparable non-human predators (birds of prey, bony and cartilaginous fishes, terrestrial mammalian carnivores) with extensive ranges and broad dietary niches comprised primarily of other vertebrates as prey. Publications that provide comprehensive and range-wide lists of dietary items at the species level of prey are rare. We started with papers that we knew contained range-wide dietary data, which included seven papers on terrestrial carnivores (data repository[43]). We then either downloaded[44] or asked authors for data[45], requesting increased taxonomic resolution compared with original papers where necessary[46]. We also secured range-wide dietary data from S. Birrer on two owl species, as well as from J. Ford on killer whales (*Orcinus orca*). From a total of 2,779 prey items originally identified at the species, genus, or family level across 19 predators for which IUCN range data also exist, we identified 1,958 unique vertebrate prey species (2455 when species across predators are counted more than once). These are prey that could be resolved to the species level by original authors or via our validation methods (below) and appeared in the IUCN database. These species formed the basis of our comparisons of prey diversity and dietary overlap between humans and other predators of vertebrates.

No recent, globally relevant dietary summary existed for any cartilaginous fish, so we created one for white sharks (*Carcharodon carcharias*). We started by summarizing vertebrate prey species identified in a previous global review[47] ($n = 63$; 4 of which resolved only to genus). We used 'white shark OR *Carcharodon carcharias*' AND 'diet' as search terms within titles, abstracts, and text on Google Scholar, JSTOR, and Web of Science to identify potentially relevant data published since 1999. Returns totaled 913 papers, which we sorted via 'relevance' (automated sorting features provided by each search engine). We inspected 127 papers that had abstracts indicating the likely availability of data (in the form of behavioral observation or stomach content analysis). Cross-referencing with items identified in the global review[47], we extracted additional vertebrate prey items ($n = 71$; 8 of which resolved only to genus level) from 25 papers.

We sought confirmation that names of vertebrate species identified in predator dietary studies were accurate and listed in the IUCN database, attempting to resolve inconsistencies. Given taxonomic changes between when original data were recorded or published and the IUCN data we extracted, as well as potential data entry errors, we made efforts to detect cryptic matches. Using the genus/species binomials of prey, we searched for matches within Chordata on our version of the IUCN database using a string-based query. We accounted for potential divergence in matching binomials (e.g., differences in spelling) based on optimal string alignment using the R package *stringdist*[48]. Upon inspection, string distances between binomials greater than 0.9 were in all cases owing from differences in spelling and easily resolved. Those with distances greater than 0.5 were investigated as potential matches using the following steps, with two potential outcomes: (1) we queried the Integrated Taxonomic Information System database[49], with the binomial to investigate recommendations made by this tool. If no reasonable matches were found, the prey species was listed as 'unknown' and not counted as a species in diversity comparison or dietary overlap metrics described below; (2) if resolved, the validated prey binomials were queried again using the IUCN database; only matches were retained and further considered.

Using this cleaned dataset, we compared the number of vertebrate species preyed on by each comparator predator with the number of species used by humans over each predator's distribution (as estimated by species range maps; detailed mapping methods below). Specifically, we calculated the prey diversity (i.e., number of prey species) of comparator predators and of humans (we quantified prey diversity of humans based on 'all uses' [Supplementary Table 1] and for 'food'

[use as food for humans] only), as well as the difference in prey diversity between the comparator predators and humans (e.g., 2707 human prey species/9 jaguar prey species = ~300 times more species). We also estimated dietary overlap by calculating the percent of a comparator predator's prey species that are also used by humans.

Several limitations are relevant to the coarse scale at which data occur and at which we compute these estimates and make comparisons between humans and other predators. First, the studies upon which we drew surely missed species used by comparator predators. Also, when querying the IUCN database for species used by humans, those identified might only occur in a modest proportion of the predator's range, thereby limiting opportunities for (documented) predation. On the other hand, although humans are perhaps the best studied species on the planet, the IUCN database likely fails to record human uses of assessed species (above), leading to an underestimate of prey used by humans. Additionally, all comparator predators might also use invertebrate prey. However, our comparisons relate to vertebrates only.

**Mapping**. To display global variation in use of vertebrates by humans (and in the 'comparisons with other predators' analysis), we took several steps. We used 2020 spatial data from IUCN[39] for all vertebrate classes except birds, for which we used 2019 BirdLife International data[50]; both sources represent species' ranges as polygons (or HydroBasin polygons in the case of some freshwater species) or points[50]. First, we matched the species range polygons, or points if no range polygons were available ($n = 886$), with the IUCN data based on corresponding taxonIDs (individual identifying codes used by IUCN). In the case of HydroBasins, we matched taxonIDs with the corresponding HydroBasinID. We then queried our IUCN database to determine whether there were additional species that matched the range data based on the binomial and not the taxonID. In total, spatial data were available for 40,096 species (of 46,755; 86%). We intersected matched range polygons with a 110 km² Plate Carrée global grid using GeoPandas[51], creating a unique presence record for each species in a grid cell with associated IUCN use data. We used the grid intersections to map human use by tabulating the number of unique species with any human use, use as food, and use as pets in each grid. Additionally, we fit a negative binomial generalized linear model (*glm.nb* function; MASS package[52]) of used species over log transformed available species, using each cell as a case (Fig. 5) and mapped the residuals. Residual values thus identify cells in which more or less species are used by humans as predicted by generalized (linear) relationships between use and availability. Outliers in the residuals generally corresponded with polar regions, where there were either no recorded exploited species (e.g., Antarctica) or where species diversity was low and the proportion of species exploited was high (e.g., the Arctic).

**Trait data**. For the trait analysis, we focused on birds and mammals (16,413 species), as trait coverage is poor for other vertebrate classes (e.g., reptiles, amphibians[53]), or relevant traits are incomparable (e.g., fish[54]) (i.e., separate trait spaces could be created for each class[55] but this prohibits comparisons across taxa). We also filtered to only terrestrial species (based on IUCN realm definitions), due to differences in trait-environment relationships for exclusively aquatic species[56,57]. We matched the trait data to the use data using species binomials, as both were constructed based on the IUCN taxonomy.

We used five traits: body mass, litter/clutch size, habitat breadth (number of IUCN habitats listed as suitable), generation length and diet (the dominant diet gradient across seven diet categories for all species, see below) that have previously been used to summarize bird and mammal ecological strategies[58]. We extracted raw trait data for body mass, litter/clutch size, habitat breadth, and diet from a trait database for birds and mammals[58] (previously compiled from four main sources[59-62]). We used published generation length values for birds[63] and mammals[59].

We prepared dietary data for these analyses. Specifically, raw diet information was available as semi-quantitative records (percentage use of ten different dietary categories[62]). We reclassified these to seven dietary categories (we summed the "vertebrate fish", "vertebrate endotherms", "vertebrate ectotherms" and "vertebrate unknown" categories into a "vertebrates" category). We then converted this diet information into a continuous measure of a species' diet, broadly following Cooke et al.[58], so that we could integrate diet into our analyses. To convert the diet information into a continuous measure, we extracted the first principal component from a principal coordinates analysis (PCoA) of Gower distances based on the seven dietary proportions. These values serve as synthetic trait values (i.e., new trait values based on the relative importance of diet categories in the initial dataset) and are referred to as 'diet'. Diet explained 37% of the variation across the diet categories and was predominantly loaded positively on invertebrates (PCoA loading = 3.7) and negatively on plant material (−1.7), fruit (−1.2), and seed (−0.8).

We transformed trait data where it improved homogeneity of variance: $\log_{10}$ for body mass and generation length; square root for litter/clutch size and habitat breadth; and we standardized all traits to zero mean and unit variance (*z*-transformation). Transformation and standardization to unitless coordinates is recommended for trait analyses and hypervolume calculations[64,65].

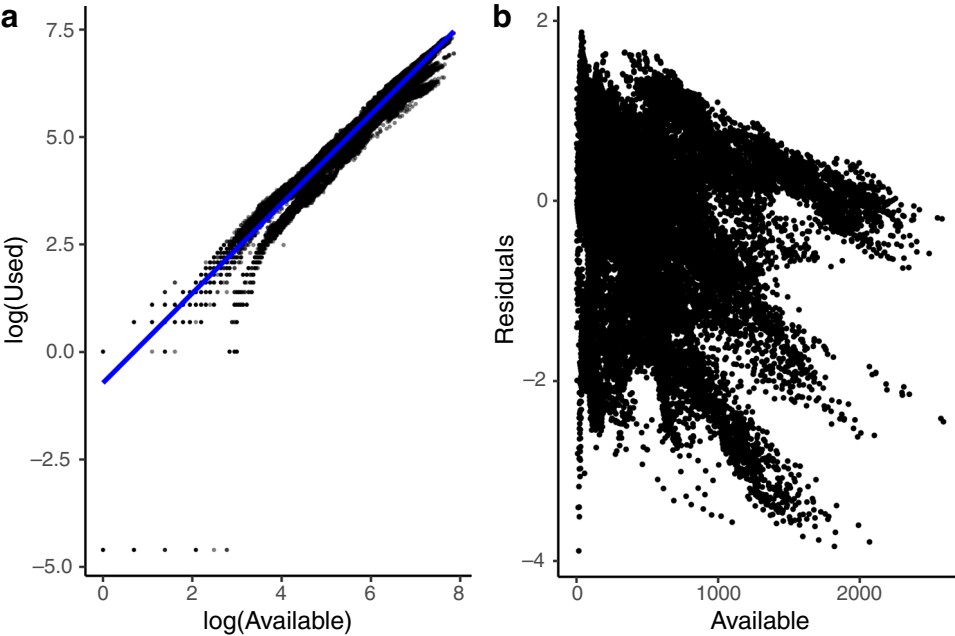

**Fig. 5 Residuals data underlying mapping in Fig. 3. a** Species used by humans ($\log(x + 0.01)$) as a function of species present (log) in each 110 km$^2$ grid cell across the planet ($n = 71{,}566$). **b** Relationship between Pearson residuals from **b** and raw counts of species present in grid cells.

**Trait imputation**. All five traits had >50% coverage across species, and overall 83% of trait values were complete. To achieve complete species trait coverage, we imputed missing data, as imputation increases sample size and, consequently, the statistical power of any analysis while reducing bias and error[66–68]. We estimated missing values using random forest regression trees[69,70], an approach that has high predictive accuracy and the capacity to deal with complexity in relationships, including nonlinearities and interactions[71]. To perform the random forest imputations, we used the *missForest* function (missForest package[71]). We imputed missing values based on the ecological (the trait data) and phylogenetic (the first 10 phylogenetic eigenvectors) relationships between species for birds and mammals separately (due to separate phylogenetic trees[72,73]). We included previously calculated phylogenetic eigenvectors (the variation in the phylogenetic distances among bird species[74] and mammal species[58]), as phylogenetic data can improve the estimation of missing trait values in the imputation process[66]. We selected the first 10 phylogenetic eigenvectors to ensure a balance between including detailed phylogenetic information and diluting the information contained in the other traits[69], where the first 10 phylogenetic eigenvectors summarize major phylogenetic differences between species[74]. To increase predictive accuracy and prevent over-fitting, we generated 500 random forest regression trees; a cautiously large number[71]. We set the number of variables randomly sampled at each split (mtry) as the square root of the number variables included (10 phylogenetic eigenvectors, five traits; mtry = 4); a compromise between imputation error and computation time[71]. To capture the imputation uncertainty, we generated 15 imputed trait datasets, which is suggested to be sufficient[69,75]. These imputed datasets are based on the same input trait data but differ in their estimations for the missing data.

To quantify the average error in the random forest predictions across the imputed datasets (out-of-bag error), we calculated the mean and standard deviation of normalized root squared error across the 15 datasets for birds (habitat breadth = 26.6 ± 0.3%; clutch size = 10.6 ± 0.2%; body mass = 7.3 ± 0.7%; diet = 7.2 ± 0.5%; generation length was complete for birds) and mammals (habitat breadth = 22.9 ± 0.1%; clutch size = 12.2 ± 0.1%; body mass = 9.5 ± 0.3%; diet = 4.8 ± 0.1%; generation length = 4.7 ± 0.1%)[62]. Low imputation accuracy is reflected in high out-of-bag error values, where habitat breadth had the lowest imputation accuracy for both birds and mammals.

**Ecological trait space**. We built an ecological trait space from the imputed traits via principal component analysis (PCA), for each of the 15 imputed trait datasets, using the *princomp* function (vegan package[76]). We summarized the mean principal component values per species across the imputed datasets.

We then used multivariate kernel density estimation to calculate probability contours across the ecological trait space, via the *kde* function (ks package[77]). We extracted 50% probability contours for each group (i.e., 'Not used', 'Used', 'Extinction risk & use considered threat'). As results depend on the choice of the bandwidth used for the smoothing kernel, we used an unconstrained bandwidth selector[78]—the sum of asymptotic mean squared error pilot bandwidth selector[78,79]—through the *Hpi* function (ks package[77]).

We also constructed hypervolumes to assess the volume and unique regions of trait space occupied by terrestrial birds and mammals. We calculated hypervolumes using the one-class support vector machine (SVM) estimation method[80]. SVM provides a smooth fit around data that is insensitive to outliers, yields a binary boundary classification ('in' or 'out'), and is computationally viable in large datasets and high-dimensional hyperspaces[80]. We built the hypervolumes based on the mean (across the imputed datasets) principal components above using the *hypervolume_svm* function (hypervolume package[76]). The units of the hypervolumes are reported as the standard deviations of the principal components, raised to the power of the number of dimensions (SD$^5$). We constructed observed hypervolumes for each use group (i.e., 'Used', 'Extinction risk & use considered threat'). We then contrasted the observed hypervolumes with random hypervolumes. We generated 333 random hypervolumes (a balance between precision and computational demands), based on rarefied (i.e., equivalent number of species) random samples from the global pool of species. From these observed and random hypervolumes we calculated the total volume (SD$^5$) of the hypervolumes, as well as the percent unique volume. We quantified percent unique volume by intersecting the observed hypervolume and each random hypervolume, using the *hypervolume_set* function (hypervolume package[81]), and then calculating the fraction unique volume of the intersection (i.e., the regions of ecological trait space not shared by the observed and random hypervolumes), with the *hypervolume_overlap_statistics* function (hypervolume package[80]). We then divided the fraction unique volume by the total volume, times 100, to get the percent unique volume. As this percent unique volume measure is based on intersecting the observed and random hypervolumes, the observed value varies, depending on the random hypervolume's size and position in ecological trait space (hence the uncertainty associated with the observed estimates of percent unique volume). We used permutation tests (one-sided) to test for statistical differences between the observed and random hypervolumes. Specifically, we assessed whether the observed values (for both volume and percent unique volume) were greater than the random values, with: (number of random values > observed value + 1)/(number of random samples +1)[82].

To understand the differences in the traits between those birds and mammals used by humans and those not used by humans we visualized the individual trait distributions, with 95% confidence intervals reflecting the uncertainty captured by the multiple imputed datasets. We also calculated Hedge's g relative effect sizes to assess the magnitude of difference between the distributions (effsize package[83]), and used Mann–Whitney $U$-tests (two-sided) to identify statistically different distributions[70]. We performed these analyses per imputed dataset and then summarized the results with the mean across the datasets.

We also built hypervolumes based on the prey diversity (i.e., prey species) of comparator predators and of humans across equivalent geographic ranges. Of the comparator predators for which we had data (19 species), we selected only those whose prey species were primarily (>80%) terrestrial birds and mammals (14 comparator predators)—reflecting the available trait data. We matched the prey species to the prepared trait data by their scientific names for each comparator predator and for humans. We used human prey species based on all uses, as food use was generally indistinct in volume compared to all uses, due to the non-linear scaling of volume with

number of species. We then constructed hypervolumes for each comparator predator and for humans, with the *hypervolume_svm* function (hypervolume package[81]).

**Reporting summary**. Further information on research design is available in the Nature Portfolio Reporting Summary linked to this article.

## Data availability
Data are available at https://doi.org/10.5281/zenodo.7644514.

## Code availability
Annotated code is available at https://doi.org/10.5281/zenodo.7644514.

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

## Acknowledgements
We thank Julian Ehlers and Allan Roberts for database creation and initial data visualization, as well as the Applied Conservation Science lab at the University of Victoria for constructive feedback and discussion. Additionally, we thank two reviewers for thoughtful and candid feedback that greatly improved the manuscript.

## Author contributions
Conceptualization: C.D., R.C., M.B., H.B., S.C., J.E., M.G., T.L., I.M., J.M., P.P., B.W. Data acquisition: C.D., R.C., M.B., H.B., J.M. Methodology: C.D., R.C., M.B., H.B., J.E., T.L., B.W. Analysis: R.C., M.B., C.D. Visualization: R.C., M.B. Data curation: R.C., M.B. Funding acquisition: C.D. Project administration: C.D. Writing: original draft: C.D., R.C., M.B. Writing: review & editing: C.D., R.C., M.B., H.B., S.C., J.E., M.G., T.L., I.M., J.M., P.P., B.W.

## Competing interests
The authors declare no competing interests
