## [Peer Review File · Communications Biology]

Reviewers' comments:

Reviewer #1 (Remarks to the Author):

Dear Chris and team

This is clearly an excellently conducted conservation article that also presents some intriguing analyses of human predation. The writing is engaging and I expect the media and social media will love this. It is precisely because I think this study is so well done, and so likely to be impactful, that I have shared such strong comments. It is also because I have been fortunate to be a collaborator with some of you, and that I respect your expertise and creative abilities, that I have allowed myself to express some difficult thoughts.

My review of your paper is not usual. It expresses some deep concerns I have about the moral leanings of conservation and the way it could be captured. Had I read this a mere few years ago I'd be cheering you on. Today, I am worried. Let me explain.

Our society seems more and more inclined to describing humanity as a "pest", an "invasive species" - as people become convinced by 'professionals' like us. Studies like this affect people. They affect children. They also affect those in power - but not necessarily as we would anticipate.

You will see my exact comments alongside your text. I wrote as I read, bluntly at times - forgive me. I point out examples where I consider your framing to align with and reinforce conservation's hubris, misanthropy, and even a tendency toward authoritarianism. ANY study that uses Red List opinions as if it is data will risk hubris. ANY study that starts with a question like "we set out to find out what harms humans cause" will have risk misanthropy and authoritarianism. Your study reminds me a lot of studies about feral cats. "Just look at how many birds they eat, that's why we need to control them".

I worry about the danger of continuing to treat human beings as a categorical threat to the planet. I worry about isolating a class of people as the exception that proves the rule - those few good humans defined as "indigenous" and "place-based" - who don't "exploit", they "harvest". I worry about how a study like this one could play into the hands of powerful entities that will use any excuse to control human behaviour.

I'm not asking you to make any major changes actually. This is ultimately up to you. It is an excellent paper. (though def need to acknowledge that the Red List opinions is not actually data - maybe you did - I was lazy and trusting and didn't read the methods - sorry about that too :))

Warm regards
Arian Wallach

Reviewer #2 (Remarks to the Author):

I found this to be a very interesting paper to review. And the efforts of the authors to mine the IUCN datasets so as to access these types of relationships is impressive. I am supportive of this paper, but I recommend that the authors think critically about the framing of the paper with respect to the terms of 'use' and 'predation' as I outline below in my review.

The review would have been more straightforward with line numbers. Nevertheless, I will try to convey the recommended changes.

- Not sure if I understand the framing of the study exactly. The issue lies in what the authors mean by 'six most speciose classes' (page 4). I believe that these are later referred to, in the next paragraph, as the 'six largest vertebrate classes.' Those two things don't quite mean the same thing but, unless I am mistaken, are being used interchangeably. I would recommend that the authors review the manuscript to sort out inadvertent inconsistencies such as these. Additionally, I recognize that words are limited with this submission and there is a linkage to the Methods herein, but it would be helpful to articulate to the reader why, precisely, these six classes were specifically

selected. Perhaps one more sentence to frame that would be helpful here?

- With respect to frequencies of species selection in Fig. 1, is it anticipated that there is any evident bias based on the availability of species in the IUCN database? For instance, most species in the database are presently data deficient to determine their Red List status. Given that this study examined the conservation assessment data, did the authors lose diversity in the database because of data deficiencies by IUCN?
- “Compared with other wide-ranging predators (predatory fishes, sharks, avian and mammalian predators), humans exploit many more vertebrate species.” Is this a fair comparison given that non-predation harvest (i.e., “We consider predation by humans broadly – and from the perspective of prey – as any use that removes individuals from wild populations, lethally or otherwise, via processes ranging from local subsistence to global commercial harvesting and trade.”) was included?
- “Equatorial regions, where species richness is highest, particularly coastal areas and across Southeast Asia, show the highest number of exploited species (Fig. 3A).” Biodiversity is obviously highest in the tropics, but analysing current range maps doesn’t account for the historic defaunation that occurred in higher latitudes. Not suggesting that the authors need to make changes to the analysis in line with this comment, but I encourage them to be careful regarding the interpretations of the results. I see from reading the Methods section that the authors focused on IUCN’s extant species.
- The authors experience some issues with tense throughout the manuscript. I would recommend a review with fresh eyes to locate these things where they occur. For instance, page 10 have way through the paragraph. “We find that humans target...” should be “We found that...”
- The authors seem to insert the words ‘use’ and ‘predation’ in the same spots. Use is not equivalent to predation. Rather, predation is a very specific type of use. The authors are really modeling harvest, which is inclusive of use and predation. I raise this point in line with the framing of the paper including the title (“Humanity’s diverse predatory niche”).
- Page 10: ‘likely extraordinary’ comes off as subjective and unsupported. I would encourage a revision here.
- Discussion – The authors comprehensive assessment ‘revealed’ – Again, watch for tense.
- And once again, the outsized predatory niche is that large because it includes things that are beyond the scope of predation. I appreciate the that the authors acknowledge some of this on page 13, but I am wondering how large the predatory niche would be if it only focused on actually predation, rather than other forms of exploitation.
- Second paragraph on page 12. This introductory sentence might read better if it wasn’t interrogative.
- Top of page 15 – “losses likely to further restructure...” This reads awkwardly and is in need of revision.
- Page 16, last paragraph. The estimate may not be conservative in terms of the definition of predation used herein.
- Page 19 – the way in which the authors identified the comparative predatory species seems reasonable and well-explained.
- Page 22 – I also appreciate the authors acknowledging the weaknesses of this study.
- Mapping – this section would benefit from converting from passive to active voice, as much as possible, so that the reader can track specifically what the authors have done in this study rather than which others (e.g., IUCN) has done.
- There is a mistaken space between citations 50 and 51.

Below please see our point-by-point responses to the Referees' comments. Line and page numbers refer to the marked-up (i.e., Track Changes) version of the manuscript. We will also upload a clean copy with all Track Changes accepted. Our responses below are in **blue font** and, in case font colour disappears, also start and end with *******

REVIEWER 1

R1.1. This is clearly an excellently conducted conservation article that also presents some intriguing analyses of human predation. The writing is engaging and I expect the media and social media will love this. It is precisely because I think this study is so well done, and so likely to be impactful, that I have shared such strong comments. It is also because I have been fortunate to be a collaborator with some of you, and that I respect your expertise and creative abilities, that I have allowed myself to express some difficult thoughts.

*****Thank you. You have our interest and attention. We appreciate your radical candour, and have thought openly, and responded with some significant changes in phrasing as well as some candid thoughts of our own.*****

R1.2. My review of your paper is not usual. It expresses some deep concerns I have about the moral leanings of conservation and the way it could be captured. Had I read this a mere few years ago I'd be cheering you on. Today, I am worried. Let me explain.

Our society seems more and more inclined to describing humanity as a "pest", an "invasive species" - as people become convinced by 'professionals' like us. Studies like this affect people. They affect children. They also affect those in power - but not necessarily as we would anticipate.

You will see my exact comments alongside your text. I wrote as I read, bluntly at times - forgive me. I point out examples where I consider your framing to align with and reinforce conservation's hubris, misanthropy, and even a tendency toward authoritarianism.

*****Thank you. We have focussed on your important point here: “your framing to align with and reinforce conservation's hubris, misanthropy, and even a tendency toward authoritarianism”. We have altered our language at most of the spots on the PDF at which you raise concerns in embedded comments. For example, to avoid the unnecessarily broad term ‘unsustainable exploitation’ in favour of a more precise term to describe our methods, we have changed:**

“...draw on ecological trait data of terrestrial bird and mammalian prey to identify the potential outcomes of unsustainable exploitation on the functioning of ecosystems.”

to

“...draw on ecological trait data of terrestrial bird and mammalian prey to identify the potential outcomes of losing overexploited species in terms of the ecological diversity present in ecosystems.” Line 15, page 4***

R1.3. ANY study that uses Red List opinions as if it is data will risk hubris. ANY study that starts with a question like "we set out to find out what harms humans cause" will have risk misanthropy and authoritarianism. Your study reminds me a lot of studies about feral cats. "Just look at how many birds they eat, that's why we need to control them".

Please see below in our responses to your comments embedded in the marked-up PDF. We have now carefully acknowledged the reality that the data are from expert knowledge and judgement, not always empirically-measured dimensions of sustainability. We have also acknowledged the reality of biases, limitations, etc of IUCN data. This content is detailed below in response to your PDF comment in which this concern is raised with more detail (i.e., please see below).

R1.4. I worry about the danger of continuing to treat human beings as a categorical threat to the planet. I worry about isolating a class of people as the exception that proves the rule - those few good humans defined as "indigenous" and "place-based" - who don't "exploit", they "harvest". I worry about how a study like this one could play into the hands of powerful entities that will use any excuse to control human behaviour.

***We are confused because the first two sentences in this comment seem to be at odds. On one hand, you seem to suggest that we are considering humans as a categorical threat (i.e., unambiguously and uniformly) and on the other hand you seem to suggest that we are making an inappropriate exception (and counterfactual example) for one subset of people. Our concluding message in the Discussion is much more of the latter than former. Nonetheless, we have made changes to provide increased nuance. Our patterns (and summaries thereof) reveal how exploitation by humans threatens many (but not nearly all) vertebrates and the ecological interactions associated with those threatened species. Rather than end with such a doom-and-gloom summary alone, however, we turn to both a detailed example of one system and reference another, multi-system empirical study to illustrate how in particular some Indigenous, place-based societies have governance arrangements that are associated with sustainable exploitation over very long time periods. In the detailed responses to

your PDF comments below, we have made some modifications, which hopefully makes this important nuance clearer (please see below).***

R1.5. I'm not asking you to make any major changes actually. This is ultimately up to you. It is an excellent paper. (though def need to acknowledge that the Red List opinions is not actually data - maybe you did –

Addressed. As you suggested, we now in detail point out the limitations of the Red List assessments. Plus, we have made changes to phrasing etc related to a large proportion of your comments. So please know that we have made extensive efforts to address your thoughtful (and challenging!) suggestions.

BELOW WE ADDRESS REVIEWER 1'S COMMENTS/SUGGESTIONS ON THE MANUSCRIPT PDF. PHRASES HIGHLIGHTED BY REVIEWER 1 ARE PULLED OUT IN PARANTHESES, WITH REVIEWER COMMENTS/SUGGESTIONS FOLLOWING. OUR RESPONSES FOLLOW A LINE SPACE AND ***

R1.6 [Re: "...ii) compare humanity's predatory niche to other widespread predators] What I like about this question is that it allows room for curiosity and open mindedness, not only for the writer but for the reader

Thank you. This comment helps us understand your following point better too.

R1.7. [Re: "We also: i) ask how these interactions threaten prey..."]. Setting out the question in this format essentially negates the need for a study. You've already decided that human predators are harmful. Are there no mutualistic/facilitative effects of the human predator in the same way that other predators are valued? Conservation, in my view, has a tendency toward misanthropy and authoritarianism which a question like this reinforces.

***Although there is a large literature to support the idea that humans commonly threaten prey populations, we take your point that our specific approach/question has never before been undertaken. So, we have now included a 'might' in the phrase to allow room for uncertainty at the beginning of the manuscript before data are presented.

That is a super fascinating idea that human predators, like other predators, might have mutualistic/facilitative effects. Widespread killing of wolves by humans, for example, has facilitated larger distributions and higher densities by coyotes. While this is fertile ground to examine in follow up work, we resist the temptation to add content here.

As far as pulling back on phrases that have misanthropic or authoritarian leanings, please see some examples below in which we have strived to find more neutral or matter-of-fact terms to describe what we studied, the results, and their implications.***

R1.8. [Re: "...unsustainable exploitation"]. Here I wonder - are you asserting at the outset that human predators are unsustainable or does this allow for human predation that is "sustainable" (whatever this word means).

***Please see below. We have now crisply defined what we mean by 'sustainable' and 'exploitation'. Additionally, we now state explicitly in the Discussion that for most (i.e., 2/3rds) species, exploitation is not considered a threat. Not that such a result compensates for the staggeringly large proportion and number of species for which this is not the case, but both can be true and we should not overemphasize the gloom without acknowledging the brighter reality. Accordingly, we have now added the following detail-rich nuance to the Discussion's opening paragraph:

"On the other hand, despite a high proportion and enormous number of species considered by the IUCN as threatened by exploitation, for many vertebrates (e.g., most ray-finned fishes) harvests are not considered a threat to populations (Fig. 1B). Moreover, many species can face more severe threats from other human activities, namely habitat destruction, invasive species, and climate change. Notably, however, in a recent, taxonomically-broad and global assessment, Jaureguiberry et al (15) found that direct exploitation and land/sea use change were dominant drivers of biodiversity loss." Line 4, page 18***

R1.9. [Re: "Exploitation"] Why this loaded word? Is everything humans do to nourish themselves "exploitation"? And if so, should we use this type of language to describe all predators and other consumers?

We agree that, like many words, it is a loaded term. On the other hand, it's perhaps the most common word used in the related literature (e.g., IPBES 2019; Ripple et al. 2019; Scheffers et al. 2019). Indeed, the 'sustainable exploitation' paradigm in the resource management literature is prominent. Anyways, despite trying, we fail to come up with a better word.

R1.10. [Re: "functioning of ecosystems"] This means anything, everything, and nothing at all

***We have modified to, "Confronting the losses of species and the associated variation in ecological strategies present in ecosystems. line 13, page 21. ***

R1.11. So does "overexploitation" = "unsustainable exploitation" and what do either actually mean?

Addressed. We have now added, “Red List data are ultimately derived from expert knowledge, in combination with empirical data where available. Here, we consider assessments of species that both: i) identify exploitation as a threat and ii) list them as at risk of extinction (i.e., Vulnerable, Endangered, or Critically Endangered IUCN Red list status) as a signal of overexploitation.” Line 5, page 6.

R1.12. [Re: “From the perspectives of prey...”] Would be cool if we could ask them, but I assume you mean something else... :)

***Good point. This now reads, “from the perspective of effects on prey populations”. Line 9, page 5. ***

R1.13. I don't know if you deal with this somewhere in the methods (I'm just commenting as I'm reading) but do you account for, or at least acknowledge, the biases and ideologies intrinsic to the Red List? The Red List isn't "data" per se - although it is often treated as if it is. It is compiled by people with privileged access who hold particular values, cultures, and ideological commitments. I'm willing to wager that not many of its contributors are hunters...

Note that the Red List excludes species defined as "non native", some of which are important human prey. In Australia, wild boar, wild goat, rabbits, hares, wild cattle, wild camels, wild cats, several introduced fishes etc etc are major prey

***A good point. Although we have now included in the Introduction a statement about how IUCN assessment data are derived from expert knowledge, we agree that a recognition about those who provide such assessments should be included. So as to not distract too much from the flow, however, we have placed this statement in the Methods. We now add,

“We acknowledge that although empirical evidence often underlies IUCN assessments, they are completed by people with privileged access to the evaluation process who hold particular values, cultures, and ideological commitments. Moreover, those who assess do not declare whether they hunt or fish the species they assess.” Line 3, page 23. ***

R1.14. [Re: “For the same set of ~47,000 vertebrate species...”] I don't understand. Same as what? And how were vertebrates selected?

***Clarified. It now reads, “To capture this breadth of activities, we collated IUCN ‘use and trade’ data, which categorize the uses for which species are killed (e.g., for human food, animal food, sport hunting/specimen collection) or collected from the wild (e.g., for use as pets, for establishing *ex-situ* production) for 46,755 vertebrate species; across the six vertebrate classes with the most species (i.e., excluding classes with < 100

species), we also examined IUCN assessment data for every listed species (i.e., threats, Red List status; see *Describing Uses & Threats* in Methods)...” line 19, page 5.

R1.15. [Re: six most speciose classes] Which classes? Why did you choose those in particular? I see the "details in Methods" bit but important to understand how the set of species were selected.

Done. Please see directly above

R1.16. [Re: “Across the six largest vertebrate classes, humans use almost one third of species”]. Wowsa! Really?? Doesn't sound very appetizing to me... :) This is where it would help to understand which "species" are these? If it was 33% of bovines I wouldn't be surprised, but 33% of all birds would be very surprising indeed (thinking of people eating fairy wrens and hummingbirds makes me rather sad... :))

***We think that the word 'largest' might be unclear. It was a synonym for 'speciose' in the last sentence, which also caused confusion. We meant the classes with the most species in them (not the largest body size). Accordingly, we have modified the phrase to read,

“Humans use roughly a third of the species across the six vertebrate classes we examined (n = 14,663 of 46,755; 31%; Fig. 1A)” line 13, page 6***

R1.17. [Re: “...43% of assessed taxa”]. Again how were they selected for assessment?

We hope the modifications mentioned directly above provide clarity

R1.18. [Re: “...and birds (46%)”]. OK no way! Which birds are you looking at? I'm sure you explain this very clearly in the methods but this article format does force you to include more information in the prose.

***OK. We are better understanding our lack of clarity in identifying how we focused our analyses and figures on the (minor) subset. We excluded the handful of classes that have < 100 species....so as to keep figures manageable with 6 classes on which to report. We will make it clear that within assessed classes, we assessed all species....at least those on the IUCN red list. Accordingly, we now add,

“... we also examined IUCN assessment data for every listed species ...” line 4, page 6.

R1.19. [Re: "...considered a threat"]. It needs to be acknowledged that the Red List does not require for there to be strong evidence that something is a threat to list it so. It is almost always "expert opinion", and expert opinion is prone to biases. It is a problem when the opinions of a select few is magically turned into data which then mutates into graphs (lovely graphs BTW!), which then evolves into complex analyses, and voila published in fancy shmancy journals...

Done. As noted above, we now include a sentence about the privilege, biases, ideologies, etc of those who provide assessments. Also, we note that we always use the term 'considered a threat' (as opposed to 'is/poses a threat') throughout the manuscript, so as to remind readers that the assessments are made via knowledge and associated judgement.

R1.20. [Re: "comparable predators"]. Comparable in what sense? How were these selected? And I notice that most of these are obligate predators while humans are omnivores and worldwide humans primarily eat plants - so I'd think that bears, boars, various apes and monkeys etc would be more similar.

***We define 'comparable' in two paragraphs after this figure caption, stating, "...Compared with other wide-ranging predators of vertebrates (i.e., predatory fishes, sharks, avian and mammalian predators)'. It's unfortunate that the figure necessarily comes before the text (at least prior to formatting for potential publication). So, we have added "(i.e., those that prey on vertebrates)" to the figure caption.

We acknowledge that humans are indeed omnivores but we reasoned that the most stringent/conservative comparisons would be with obligate predators that we assumed would consume more prey species than omnivores like pigs, bears, etc. For this reason, we retain this set of comparators (the data for which took us over a year to collect).***

R1.20. [Re: all vertebrates]. So you did assess all vertebrates the Red List has data for?

Yes (minus those several hundred out of ~47,000 in the classes in which there are few species). Hopefully, this is now clear from changes detailed above

R1.21. [Re: contributing threat']. You mean there are two categories "threat" and "contributing threat"? Where is this from? At least the species I've looked at are not ranked. Again, the reliability of such rankings is probably very very low

Done. We changed to 'a threat', because we have not assessed how severe human 'use' is as a threat compared to, say, climate change and habitat destruction. (We have, however, now acknowledged the frequency and severity of those two other threats in the opening part of the Discussion). Moreover, we agree that threats are considered so based on judgement, and we have hopefully addressed this concern above

R1.22. [Re: “compared with other wide-ranging predators (predatory fishes, sharks, avian and mammalian predators)] Not sure what you mean.

***Done. We have added another phrase and an all important ‘ie’ to provide more clarity here, modifying to: “Compared with other wide-ranging predators of vertebrates (i.e., predatory fishes, sharks, avian and mammalian predators)”. Line 10, page 8. ***

R1.23. [Re: “Paired comparisons over equivalent geographic ranges with 19 vertebrate predators, for which range-wide dietary data exist (Methods), reveal that humans prey on ~5 to ~300 times more vertebrate species]. But again, how do you account for human omnivory and high dependence on meat from domestic animals. That is what is actually the % in the diet of humans of all these wild vertebrates?

***Addressed. We mean that humans prey on 5 (on the low end) to 300 (on the high end) times the number of vertebrate prey species than the predators to which we compare humans. For (the high end) example, across the jaguar’s range, it kills 9 vertebrate species. Assessed over than same jaguar range, humans exploit 9 x 300 or ~2700 species (actually, 2,707 as read in Figure 1C). We provide details on these calculations in the Methods: “... (e.g., 2,707 human prey species / 9 jaguar prey species = ~300 times more species)”. Line 8, page 29

So as to not conflate with proportion of diet or other measures, we now use the word ‘number’. Such comparisons need not account analytically for other forms of sustenance, like domesticated animals (though such ‘subsidy’ from domesticated plants and animals in part helps to explain why humans can maintain such a broad predatory niche and often have high exploitation rates). Anyways, we have modified the sentence to now read:

“Paired comparisons over equivalent geographic ranges with 19 vertebrate predators, for which range-wide dietary data exist (see *Prey diversity comparisons* in Methods), reveal that humans exploit ~5 to ~300 times the number of vertebrate species (~4 to ~122 times, considering food items only; Fig. 1C)”. line 12, page 8. ***

R1.24. [Re: “pet use is almost twice as common (74%)...] You mean food for pets or the animal is adopted as a pet? (I assume the former but just want to make sure) If it is about being adopted as pets "pet use" is odd terminology

***Addressed. ‘Pets’ strictly refers to use of animals as pets, with use as food for pets covered by animal feed (see Table ED1). We have now made this clearer in the text:

“... ranging from food for humans, pets (i.e., companions or captive animals,)...”. Line 20, page 8.

and we have changed the line in question to: "... in the terrestrial realm, use as pets is almost twice as common (74%) as food use (39%)". line 4, page 9. ***

R1.25. [Re: Figure 2.] Hmm a little confusing on the eye. Pretty but confusing. Especially where arrows cross over. Might be easier if species photos of the same category are bunched together or just have a representative photo for each graph? Anyway up to you of course, just sharing my experience as a reader

Thank you. Despite the busy-ness, we really wanted to showcase some variation in exploitation of a diverse taxonomic and geographic set of species (some of which face threats, and some of which do not). In this way, individual species are not 'forgotten' among the ~47,000 species for which we had data. Also, we hope that it's a fascinating mini-lesson on the diverse ways that diverse species are used by humans around the world, sustainable or otherwise

R1.26. [Re: Figure 3 maps] Because these colours are linked to emotions (blue is calm, red is alarm) colour coding from blue to red again limits the readers (and writer's?) freedom to think openly

***Done. We have used another common colour ramp that does not use the blue to red variation. It's a common one called, 'magma'. We paste the new figure here

New figure at line 1, page 13.***

R1.27. [Re: “We also estimated the patterns and consequences of exploitation on the diversity of ecological traits across terrestrial birds and mammals”]. The word “also” diminishes what seems to me the most interesting of your analyses. It doesn't merely preach at me - “oh look how horrible human hunters/fishers are! They eat everything!! Exploit everyone!!!” - it actually is teaching me something about the human predator - AND sets this in a conservation context

***Addressed. We like your ‘conservation context’ phrase and have modified the sentence to read, “To provide additional conservation context, which relates to ecosystem function, we also estimated the patterns and consequences of exploitation on the diversity of ecological traits across terrestrial birds and mammals (n = 16,413 species). Line 10, page 14. ***

R1.28. [Re: “Moreover, those species at risk of extinction and for which use is considered a threat occupy a disproportionately large (Fig. 4B; permutation test $P \leq 0.05$) and unique (Fig. 4C; $P \leq 0.05$) region of trait space.” I don't quite understand. These types of analyses, which are becoming more and more common, are quite demanding on the reader so require compassion from the writer. Please take us readers by the hand and explain things as simply as possible...

***In the several sentences prior, we had introduced these analyses to the reader and have now directly pointed them to the relevant section in the Methods, stating:

“...we then used principal component analysis to examine if humans exploit (and overexploit) species non-randomly across trait space, assessing the volume and uniqueness of exploited trait space (details in *Trait data to Ecological trait space* sections of Methods)”. line 15, page 14. ***

R1.29. [Re: “Humans are likely extraordinary in exerting such influence...”]. I really don't know what to make of a statement like this.

***Addressed. We had used ‘extraordinary’ in its plainest sense, distinguishing humans from other ‘ordinary’ predators (with constrained predatory niches), but we understand it likely sounds too hyperbolic. Accordingly, the phrase now reads more to the point,

“Finally, humans are seemingly unique among predators in interacting so broadly with the ecological trait space of birds and mammals” line 2, page 15. ***

R1.30. [Re: fig 4 caption; “...comparisons of (B) total volume and (C) unique volume between observed and randomized trait spaces]. Sorry I don't understand.

***We hope that the additional information provided to address the similar comment about volumes of ecological trait space provides clarity. Additionally, we have placed a “See *Ecological trait space* section of Methods for details” [line 6, page 17] signpost in the caption to assist readers. ***

R1.31. [Re: “...that of comparable predators”] This really doesn’t make sense. What makes a predator defined as comparable? What does it mean to compare a human to an owl?

***Addressed in a couple of ways. First, we have now modified the sentence to read,

“This uniquely large predatory role -- up to 300-times taxonomically and 1,300-times ecologically larger than those of the non-human predators to which we had comparable data -- is driven by a wide variety of uses, many of which are independent of sustenance”. Line 13, page 17

Second, as mentioned above, we have now better explained why we compared to other predators of vertebrates (and specifically those with range-wide dietary data).. ***

R1.32. [Re: “this outsized predatory footprint...”]. Unnecessarily ideological/misanthropic. ALSO Again ideological/misanthropic and you did not actually measure the quantity and ecological effects of human predation

***Addressed. By ‘outsized’, we meant larger than those to which we compared, but we acknowledge your point. And we used ‘footprint’ as a synonym for ‘effect’, but also acknowledge your point. Accordingly, we have modified this phrase to read, “This uniquely large predatory role ...” line 13, page 17 ***

R1.33. [Re: “Use for pets, medicines, and other wildlife products, for example, are not only common but now also dominates wildlife endangerment on a global scale (Figs. 2; 3D; see also 8, 9)”] Well, human distribution is on a global scale so this just is hyperbolic

***Addressed. Whereas the global distribution of humanity does not necessarily mean human endangerment of wildlife will also be global, the reality as shown in the two studies we cite in this sentence (and more) is that the endangerment is indeed manifest on a global scale. We do, however, acknowledge your point about hyperbole (and see it in the verb ‘dominates’ we used. Accordingly, to be consistent with other changes in the manuscript tone, we have modified the sentence to,

“Use for pets, medicines, and other wildlife products, for example, are not only common (Figs. 2; 3D) but now also underlies the exploitation-related endangerment of wildlife in many areas (9)” line 17, page 17. ***

R1.34. [Re: *same sentence as above*]. This isn't evidenced. You can say that species categorized as endangered are also hunted and taken for other purposes, and that the Red List contributors believe that human predation is also a cause of endangerment. You cannot say anything about the relative effect of predation compared to other issues (you don't know that predation "dominates" anything).

Addressed. We definitely see your point here. That was not the meaning we intended. We believe the change we made directly above, however, provides clarity

R1.35. [Re: Ecologically, however, these uses can have the same effect as predation for food by removing, and in many cases extirpating, local populations."] Examples?

***Addressed. Rather than adding examples (say, about some of the large declines and extirpations of many populations collected for traditional medicine use), we have removed the 'in many cases extirpating'. To add examples, we would need to add citations, Latin names and brief context, which would distract from the flow. The sentence now reads,

"Ecologically, however, these uses can have the same effect as predation for food by removing individuals from populations". Line 3, page 19 ***

R1.36. [Re: In this way, global commerce and trade – uniquely human endeavours – also fuel the industrialized human's diverse predatory niche.]. Would be fantastic if you could distinguish between commercial-international and subsistence-local "exploitation". There is of course a need for reflection when a bunch of wealthy academics point the finger at folk hunting and fishing to feed themselves and their kin, or selling their wares to make a basic living...

***Agreed. Although the IUCN data do not allow for such detailed discrimination between subsistence and commercial use of wildlife, this is an important question to ask. Soon you encounter our final paragraph, which brings this issue to the fore. Please see below how we have modified that last paragraph (after accounting for your other comments on this topic). ***

R1.37. [Re: "subsistence" {the term}]. My understanding of the term is not about the technology employed but the use of the hunted. This is how I would use the terms: If I hunt with a rifle to feed my family I'm a "subsistence hunter". If I set snares made of natural fibre and sell my wares into the international trade, say for the Chinese medicine industry, I'm industrialised. No?

***We agree on the definition of the term. We are confused because there is no reference to technology in the paragraph. We now, however, remove the term and have given the paragraph a better introductory sentence. It now reads,

“Although detailed comparisons at standardized scales are not possible, humanity -- now industrialized -- likely has a far broader predatory niche than at any time in history. Line 11, page 19. ***

R1.38. [Re: “...which Flannery (21) deemed the ‘broad spectrum revolution’). What does this mean?

***Addressed. Although ref 21 describes it in detail, we have modified the sentence in which it appears to make it clear that this sentence in fact contains the definition. It now reads,

“On the other hand, the early diversification of the human niche proceeded with and paralleled, the development of environmental management techniques (e.g., fire), and the later advent of agriculture and animal husbandry (7), a process that Flannery (24) termed the ‘broad spectrum revolution’. Line 16, page 19. ***

R1.39. [Re: “...the planet’s most extraordinary predator]. Now I’m not sure if you’re misanthropic or a cheer leader for humanity. We’re not merely extraordinary but most extraordinary :) Personally, I think water rats are the most extraordinary predators :)

Lol. We understand. Again, we used ‘extraordinary’ not in the ‘fabulous’ sense but rather in the ‘extra-to-ordinary’ context. But this has made it clear that it potentially will be unclear to many other readers. Accordingly, we now use the term, ‘widespread’, which works in the context of both geography and prey diversity.

R1.40. [Re: Against this background of a broad niche, high mortality risk, and strong selection pressures, even the perceived threat of predation associated with benign human activity has altered the behavior of many taxa (26–28)] What do you mean? What about human shielding? What about humans befriending and feeding wild animals?

***Whereas human shielding provides a counter-example, it seems like a relatively rare phenomenon compared to the many hundreds of systems showing the changes in spatial and temporal activity patterns in wildlife, as summarized in the meta-analyses we cite (Gaynor et al. 2018; Tucker et al., 2018). And, although some humans can befriend and feed wild animals, our suspicion is that this is not common, and happens primarily if interactions start in the juvenile (and maximally social) period of development among some wildlife. For these two reasons, we feel like introducing these two counter

scenarios would not add much value compared to the costs in disruption to flow from explanations, citations and so on.***

R1.41. [Re: same above sentence]. Humans can't alter the behaviour of other animals. It is the animals that alter their own behaviour. Also the tone of this seems negative but I'm not sure why. Why should a change in behavior be considered a bad thing? Or maybe I just added a negative tone where none was meant to be?

We feel that this sentence did not have a negative tone. Although those whom we cite explain how these altered spatial and temporal activity patterns in the presence of (and with abundant evidence that it is very likely caused by) human activity can indeed have negative effects on both the focal wildlife (i.e., by contracting foraging times or places) and ecosystems (nutrient cycling and disease transmission).

R1.42. [Re: Should exploitation be as taxonomically widespread as our data suggest]. Hmm... Is this actually data? That is, is the Red List - a compilation of expert opinions - "data"? You could say "Should.... be as widespread as the contributors to the Red List believe"

***Addressed. Rather than fit in a perhaps unnecessarily critical definition of Red List assessments (which the verb 'believe' might impose), we have modified the sentence to use the term 'patterns'. In other areas, and following suggestions, we have made it clear that the Red List data are expert judgements, though subject to various biases, etc. We have made the change and also note here that the sentence itself is one that is careful in its construction in how it notes,

'Should exploitation be as taxonomically widespread as the patterns we present suggest...'. Line 16, page 20. ***

R1.43. [Re: "landscapes of fear"] Keep in mind not everyone will be familiar with that term.

***Addressed. We have now added a key reference (Zanette & Clinchy 2019) and also used the broader term, 'ecology of fear'. ***

R1.44. [Re: Although affecting only a moderate proportion of all species used, the enormous number of species exploited unsustainably"] Based on what. Defined how? Examples? ALSO [additional comment on same sentence] (a) what does "ecological loss" mean? (b) how does Fig 4 evidence this?

***All three of these comments are addressed by changing the sentence to,

“Although affecting only a moderate proportion of all vertebrates used, the large number of species for which exploitation is considered a threat might likely contribute to continued loss of species (Fig. 1B; those species facing extinction risk) as well as loss of variation in ecological trait space (Fig. 4A-C). line 1, page 21.” ***

R1.45. [Re: “...left unchecked”]. There’s that authoritarian bent so common in conservation....be careful!

***Agreed. We modified phrase to, “Without changes to predatory behaviour by humans...”. Line 8, page 21. ***

R1.46. [Re: “...further restructure the ecological diversity now observed...”]. What does this mean? This is more like mystical language than scientific. AND What is observed exactly?

***Addressed. We modified the sentence and also note the sentence before it was expanded to provide more detail to support clarity. The focal sentence now reads,

“...these losses are likely to further reduce the ecological diversity present among the world’s vertebrates”. Line 8, page 21. ***

R1.47. [Re: “Taxonomic losses might accumulate and expand as existing targets decline or earn protections, causing hunters, fishers, and collectors to switch to other species that are phylogenetically, morphologically, and ecologically similar (9)”]. I don’t understand this sentence. And what is an ‘existing target’?

***Addressed. As reference 9 focuses on, the idea is that as species currently overharvested decline such that they are harder to find (or, more rarely, are given protection from harvesting), harvesters then shift their targeted species to those that are similar in those three ways. To balance increased clarity with too many words (given the key reference provided), we have modified the sentence to,

“Taxonomic losses might expand as species subject to intense exploitation decline or earn protections, causing hunters, fishers, and collectors to switch to other species that are phylogenetically, morphologically, and ecologically similar (9).” Line 11, page 21. ***

R1.48. [Re: “Confronting the losses that have accumulated under existing management Paradigms”] What does this [management paradigms] mean?

***Addressed. Rather than explain the management regimes in detail, we now replace the phrase with one more central to the theme of the manuscript. It now reads,

“Confronting the potential loss of species and the associated variation in ecological strategies present in ecosystems requires...”. Line 15, page 21. ***

R1.49. [Re: “place-based societies”]. So these are the “good humans” I suppose. Who gets to be included in this special protected class? Who decides?

Addressed. For clarity, we now add “Indigenous” as an adjective to specify about whom we write. Although we do not specifically advocate for protection of Indigenous lifeways here (given that doing so is beyond the scope of our argument), we as an author team indeed recognize the distinct rights of Indigenous Peoples to maintain their traditional ways of life. For the purpose of the manuscript, however, we point out that any conservation and management processes and decisions need to not only recognize the long-term participation of Indigenous peoples in predator-prey interactions but also highlight a data-rich example of sustainable exploitation over millennia.

R1.50. [Re: “...recognizes historic and enduring interactions between place-based societies and their prey”] So now its their prey rather than exploited prey?

***Addressed. This is now changed to,

“...and prey they have exploited over millennia”***

R1.51. [Re: “..harvesters”]. Again, so now we have kinder language toward a particular human class.

Addressed. Changed to “these interactions”, which works well, given the changes to the preceding sentence.

R1.52. [Re: “Such touchstones”]. Not sure I understand.

***Changed to “Instructive case studies like this...”. Line 6, page 22. ***

R1.53. [Re: Such touchstones provide important contrasts to the centralized ‘command-and-control’ approaches that focus on the population dynamics of prey’]. I don’t understand. I got the sense you’re advocating for command and control by conservation “experts” of human hunters.

***Addressed. Good catch. This was not at all our intention. In fact, we wanted to contrast the differences in natural resource governance approaches (and outcomes with respect to sustainability) between the decentralized place-based, Indigenous ‘touchstone’ we cite with the centralized command-and-control management

approaches. For increased clarity, we have now added the ‘decentralized’ adjective to make it clear it’s a contrast with the centralized management approach. We also reinforce the point that the former (at least in the touchstone example, but likely more broadly) is the more sustainable approach. So, if anything we are advocating for increased consideration of the value in decentralized approaches. The sentence now reads,

“Instructive case studies like this that illustrate the cultural underpinnings of decentralized harvest management provide important contrasts to the centralized ‘command-and-control’ approaches used in industrial exploitation that instead often focus on the population dynamics of prey.” Line 6, page 22. ***

R1.54. [Re: “Collaborations among social and natural scientists, as well as conservation practitioners, have looked to small-scale harvesters to learn how social and cultural practices can mitigate humanity’s destructive tendencies (e.g., 33).”] Common – is this noble savage stuff? AND Again, so now we have kinder language toward a particular human class

***Our sentence was not designed to elicit the concept of ‘noble savage’, which wrongly suggests that people outside of ‘civilization’ live in perfect harmony with the environment. Indeed, as we originally phrased the sentence, we in fact wrote of mitigating the destructive tendencies present within humanity (i.e., all humans). We also cited perhaps one of the better-known papers (a large empirical project with data across dozens of systems, published in Nature) among a large literature that has identified societal and management characteristics/norms/practices that are correlated with sustainable exploitation over the long term. That being said, and although perhaps beyond your comment, we have changed ‘destructive tendencies’ to ‘tendency to overexploit’, given the phrase is less values-laden. Finally, addressing your second comment on this sentence, we have now replaced the term, ‘harvesters’ with ‘these interactions’. ***

R1.55. [Re: “restoration of decentralized harvesting practices and the benefits they can manifest (33)]. Like what? Nice to hear some appreciation and respect toward human hunters. Come to think of it, would be interesting if you included a hunter as a coauthor...

***The benefits we were referring to relate to the decentralized management dimension of the harvesting (i.e., not centralized command-and-control to which we contrast above). So, what we really are writing about is governance. Accordingly, we have modified the sentence to read,

“...restoration of decentralized governance systems of harvesting and the sustainability benefits they can manifest (33)...” line 10, page 22

As far as co-authors that are hunters, 5 of us (including CTD) are hunters (in addition to being scientists). ***

R1.56. [Re: "...as codified in the United Nations Sustainability Development Goals and Declaration on the Rights of Indigenous Peoples, among others.]. Hmm.... I'm not sure the UN deserves such unmitigated praise

Nor do we! Although we are unsure about how the sentence allocates any praise...

R1.57. [Re: "...align with global aspirations towards social justice...". Nice to know that these are such global aspirations... Seems naive to me though.

Agreed that aspirations can be naïve. Although as an author group we wanted to acknowledge that these social justice aspirations are codified within the UN Sustainable Development Goals

R1.58. [Re: "...society needs to fully recognize the cumulative effects that its outsized predatory niche can exert on species and ecosystems]. Back to "humans should know what a terrible animal we are" - unless you're a "place-based society"

Addressed. We have changed 'its' to 'humanity's' to make it clear that the statement describes the human species and its collective impacts, not the impacts of a subset.

REVIEWER 2

R2.1. I found this to be a very interesting paper to review. And the efforts of the authors to mine the IUCN datasets so as to access these types of relationships is impressive. I am supportive of this paper, but I recommend that the authors think critically about the framing of the paper with respect to the terms of 'use' and 'predation' as I outline below in my review.

Thank you. Please see below our detailed responses to the specific concerns you raise.

R2.2. The review would have been more straightforward with line numbers. Nevertheless, I will try to convey the recommended changes.

Sorry. We now cite line numbers of revised text in the Track Changes marked-up version of the manuscript

R2.3. Not sure if I understand the framing of the study exactly. The issue lies in what the authors mean by ‘six most speciose classes’ (page 4). I believe that these are later referred to, in the next paragraph, as the ‘six largest vertebrate classes.’ Those two things don’t quite mean the same thing but, unless I am mistaken, are being used interchangeably. I would recommend that the authors review the manuscript to sort out inadvertent inconsistencies such as these. Additionally, I recognize that words are limited with this submission and there is a linkage to the Methods herein, but it would be helpful to articulate to the reader why, precisely, these six classes were specifically selected. Perhaps one more sentence to frame that would be helpful here?

***Addressed. As noted for the same concern raised by Reviewer 1, we have now more crisply and unambiguously defined how we created the subset of six classes of vertebrates on which we focus. It was always in the detailed Methods but we have added clarity to the main portion of the manuscript, noting,

“...across the six vertebrate classes with the most species (i.e., excluding classes with < 100 species)” Line 2, page 6. ***

R2.4. With respect to frequencies of species selection in Fig. 1, is it anticipated that there is any evident bias based on the availability of species in the IUCN database? For instance, most species in the database are presently data deficient to determine their Red List status. Given that this study examined the conservation assessment data, did the authors lose diversity in the database because of data deficiencies by IUCN?

***Addressed. We now add more information to the Methods to describe some of the biases and limitations. Fortunately, for vertebrates, coverage is among the very best among all species. Moreover, we cannot conceive of any directional biases. Nonetheless, with our addition, the section now reads,

“Despite the broad definition we have used, our estimate of humanity’s predatory niche is likely conservative for several reasons. For example, not all known (or unknown) vertebrate species have Red List assessments (or are included by us; above). Additionally, there might be some biases among vertebrate classes; for example, coverage for reptiles and marine/freshwater taxa is less than for other vertebrates (40).” Line 15, page 23. ***

R2.5. “Compared with other wide-ranging predators (predatory fishes, sharks, avian

and mammalian predators), humans exploit many more vertebrate species.” Is this a fair comparison given that non-predation harvest (i.e., “We consider predation by humans broadly – and from the perspective of prey – as any use that removes individuals from wild populations, lethally or otherwise, via processes ranging from local subsistence to global commercial harvesting and trade.”) was included?

***Given our explicit definition of how we considered predation from the perspective of prey, we do think it’s a fair and interesting comparison. To make direct comparisons between prey numbers consumed only for food, however, we now bring increased attention to an important part of figure 1C. Whereas the figure itself annotates prey diversity for both food and non-food prey items for humans, we have now created a summary statement in the main text that relates to the food item comparison only. It now reads,

“Paired comparisons over equivalent geographic ranges with 19 vertebrate predators, for which range-wide dietary data exist (see *Prey diversity comparisons* in Methods), reveal that humans exploit ~5 to ~300 times the number of vertebrate species (~4 to ~122 times, considering food items only; Fig. 1C)” Line 12, page 8. ***

R2.6. “Equatorial regions, where species richness is highest, particularly coastal areas and across Southeast Asia, show the highest number of exploited species (Fig. 3A).” Biodiversity is obviously highest in the tropics, but analysing current range maps doesn’t account for the historic defaunation that occurred in higher latitudes. Not suggesting that the authors need to make changes to the analysis in line with this comment, but I encourage them to be careful regarding the interpretations of the results. I see from reading the Methods section that the authors focused on IUCN’s extant species.

***This is an important point. The reader needs to be reminded (beyond the Methods) that all our analyses are just from one snapshot in time, and do not consider any historical processes. But we can bring them up in the Discussion to add richer detail. We now add,

“We also note that our contemporary ‘snapshot’ of IUCN assessments cannot capture the exploitation-related loss of species (i.e., ‘defaunation’) that has already occurred over previous centuries (13) and millennia (14) of predation by humans.”. Line 1, page 18. ***

R2.7. The authors experience some issues with tense throughout the manuscript. I would recommend a review with fresh eyes to locate these things where they occur. For instance, page 10 have way through the paragraph. “We find that humans target...” should be “We found that...”

****Addressed.****

R2.8. The authors seem to insert the words ‘use’ and ‘predation’ in the same spots. Use is not equivalent to predation. Rather, predation is a very specific type of use. The authors are really modeling harvest, which is inclusive of use and predation. I raise this point in line with the framing of the paper including the title (“Humanity’s diverse predatory niche”).

*******We have done so purposely. As mentioned above and stated explicitly in the manuscript, we are considering predation broadly. We have updated our explanation of why we have done so in the Introduction. Although everyone might not agree with our labeling, we are now more explicit in why we have done so. We now write,

“We consider predation by humans broadly – and from the perspective of effects on prey populations – as any use that removes individuals from wild populations, lethally or otherwise. Processes we considered (as captured by IUCN ‘use and trade’ designations; below) ranged from removal of live individuals for the pet trade, to harvesting by local subsistence hunters and fishers, to globalized, commercial fishing and trade of vertebrates. We additionally considered this broad definition by reasoning that these varied activities all include processes (i.e., detection, capture, etc.) embodied by predation.” Line 9, page 5. *******

R2.9. Page 10: ‘likely extraordinary’ comes off as subjective and unsupported. I would encourage a revision here.

*******Addressed. Although we meant ‘extraordinary’ in the ‘extra-to-ordinary’, we acknowledge your point. This now reads,

“Finally, humans are seemingly unique among predators in interacting so broadly with the ecological trait space of birds and mammals”. Line 1, page 5. *******

R2.10. Discussion – The authors comprehensive assessment ‘revealed’ – Again, watch for tense.

*******Addressed.*******

R2.11. And once again, the outsized predatory niche is that large because it includes things that are beyond the scope of predation. I appreciate the that the authors

acknowledge some of this on page 13, but I am wondering how large the predatory niche would be if it only focused on actual predation, rather than other forms of exploitation.

***Good point. As mentioned, we now do add new estimates as they relate to figure 1C (comparisons with other predators) but nowhere do we provide an estimate of the purely food-related predatory niche of humans. We now do so, adding,

“Of these exploited species, only about 55% (8,037 species; 17% of total species assessed) are killed for food (see below for patterns related to other uses).” Line 16, page 6.

As to your point that the outsized predatory niche is that large because it includes things that are beyond the scope of predation...we agree wholeheartedly and think this is among the most interesting patterns we found. Indeed, we emphasize this point with an updated sentence in the Discussion, which now reads,

“This uniquely large predatory role -- up to 300-times taxonomically and 1,300-times ecologically larger than those of the non-human predators to which we had comparable data -- is driven by a wide variety of uses, many of which are independent of sustenance. Line 13, page 17. ***

R2.12. Second paragraph on page 12. This introductory sentence might read better if it wasn't interrogative.

***We thought about how we could change it but really like it. It comes at a 'hard' break between paragraphs, and so we feel that a unique way to start the paragraph works well to capture the attention of, and re-orient, the reader.

R2.13. Top of page 15 – “losses likely to further restructure...” This reads awkwardly and is in need of revision.

***Addressed. This now reads more precisely as,

“Without changes to predatory behaviour by humans, these losses are likely to further reduce the ecological diversity present among the world's vertebrates (Fig. 4; e.g., 34), with consequences for global ecosystem functioning (35, 36)” Line 7, page 21. ***

R2.14. Page 16, last paragraph. The estimate may not be conservative in terms of the definition of predation used herein.

***Addressed. We have now made it explicit that (for the reasons we provide) we consider our estimate conservative, even after accounting for our broad definition of predation. The sentence now begins, “Despite the broad definition we have used...”
Line 15, page 23. ***

R2.15. Page 19 – the way in which the authors identified the comparative predatory species seems reasonable and well-explained.

Thank you!

R2.16. Page 22 – I also appreciate the authors acknowledging the weaknesses of this study.

Thank you!

R2.17. Mapping – this section would benefit from converting from passive to active voice, as much as possible, so that the reader can track specifically what the authors have done in this study rather than which others (e.g., IUCN) has done.

Addressed. We have converted most of the remaining passive voice sentences to active voice.

R2.18. There is a mistaken space between citations 50 and 51.

Addressed

REVIEWERS' COMMENTS:

Reviewer #1 (Remarks to the Author):

Dear Chris and colleagues

Congratulations on your fine research article. Thank you for engaging thoughtfully with my previous comments.

I have no further issues to raise.

Kind regards
Arian

Reviewer #2 (Remarks to the Author):

The authors have done a fine job addressing my comments in this review and I support the publication of this manuscript.